# Multi-Omic Temporal Landscape of Plasma and Synovial Fluid-Derived Extracellular Vesicles Using an Experimental Model of Equine Osteoarthritis

**DOI:** 10.3390/ijms241914888

**Published:** 2023-10-04

**Authors:** James R. Anderson, Emily Johnson, Rosalind Jenkins, Stine Jacobsen, Daniel Green, Marie Walters, Louise Bundgaard, Bas A. C. Hausmans, Guus van den Akker, Tim J. M. Welting, Alzbeta Chabronova, Yalda A. Kharaz, Emily J. Clarke, Victoria James, Mandy J. Peffers

**Affiliations:** 1Department of Musculoskeletal and Ageing Science, Institute of Life Course and Medical Sciences, University of Liverpool, Liverpool L7 8TX, UKyalda@liv.ac.uk (Y.A.K.);; 2Computational Biology Facility, Liverpool Shared Research Facilities, Faculty of Health and Life Sciences, University of Liverpool, Liverpool L7 8TX, UK; 3CDSS Bioanalytical Facility, Liverpool Shared Research Facilities, Department Pharmacology and Therapeutics, University of Liverpool, Liverpool L7 8TX, UK; 4Department of Veterinary Clinical Sciences, University of Copenhagen, Taastrup, DK-1870 Copenhagen, Denmark; 5Laboratory for Experimental Orthopedics, Department of Orthopedic Surgery, Maastricht University, 6229 Maastricht, The Netherlands; b.housmans@mastrichtuniversity.nl (B.A.C.H.);; 6School of Veterinary Medicine and Science, University of Nottingham, Sutton Bonington, Loughborough, Nottingham LE12 5RD, UK

**Keywords:** extracellular vesicles, osteoarthritis, multi-omics

## Abstract

Extracellular vesicles (EVs) contribute to osteoarthritis pathogenesis through their release into joint tissues and synovial fluid. Synovial fluid-derived EVs have the potential to be direct biomarkers in the causal pathway of disease but also enable understanding of their role in disease progression. Utilizing a temporal model of osteoarthritis, we defined the changes in matched synovial fluid and plasma-derived EV small non-coding RNA and protein cargo using sequencing and mass spectrometry. Data exploration included time series clustering, factor analysis and gene enrichment interrogation. Chondrocyte signalling was analysed using luciferase-based transcription factor activity assays. EV protein cargo appears to be more important during osteoarthritis progression than small non-coding RNAs. Cluster analysis revealed plasma-EVs represented a time-dependent response to osteoarthritis induction associated with supramolecular complexes. Clusters for synovial fluid-derived EVs were associated with initial osteoarthritis response and represented immune/inflammatory pathways. Factor analysis for plasma-derived EVs correlated with day post-induction and were primarily composed of proteins modulating lipid metabolism. Synovial fluid-derived EVs factors represented intermediate filament and supramolecular complexes reflecting tissue repair. There was a significant interaction between time and osteoarthritis for CRE, NFkB, SRE, SRF with a trend for osteoarthritis synovial fluid-derived EVs at later time points to have a more pronounced effect.

## 1. Introduction

Osteoarthritis (OA) is the most common equine joint disease characterised by cartilage degradation and changes to other joint tissues [1], severely affecting welfare and performance leading to early retirement [2]. It results in substantial morbidity and mortality [3]. Despite the huge socioeconomic importance of OA, our understanding of the pathophysiological mechanisms involved is limited [4]. Characterised by synovitis [5], gradual destruction of the articular cartilage [6], formation of osteophytes [7] and remodelling of subchondral bone [8], its pathogenesis is complex, with age, previous joint injury, joint loading and genetics among its most important risk factors. In the horse, there are a number of risk factors for OA development including secondary joint trauma, farriery, ageing and conformation (reviewed [9]). Whilst studying equine OA is important to veterinary clinicians and researchers, there is also a prospect that the horse could be utilised as a translational model to study EV-based therapies for joint diseases. This is because this athletic species develops primary OA like humans, but provides substantially larger sample volumes. OA is characterised by an increase in cartilage extracellular matrix (ECM) degradation and a reduction in its production [10]. There are no disease-modifying therapeutics available, with OA medication offering symptomatic pain relief only. Early diagnosis is thus important, as substantial joint pathology is usually present at the time of clinical diagnosis. Thus, biomarkers of early-stage OA are actively sought, and among these, extracellular vesicles (EVs) are emerging as a potential source.

EVs are nanoparticles secreted by nearly all cells, enabling the transfer of proteins and RNA between cells and tissues [11] and are found in serum, synovial fluid, articular cartilage and joint cell culture media [12]. EV cargo is involved in cross-talk between cells within joint tissues and affects ECM turnover and inflammation [13,14], thus representing a crucial step in the regulation of OA (Appendix A). The role of EVs in OA provides a foundation to create novel disease-modifying treatments [13,15]. Promising results were obtained in the therapeutic application of mesenchymal stem cell-derived EVs for cartilage repair in experimental OA [16]. Additionally, EVs have therapeutic potential in rheumatoid arthritis [13]. Synovial fluid-derived EVs have a close relationship with the pathogenesis of arthritis [17]. Recent studies described synovial fluid-derived EVs as biomarkers for different stages of joint disease [18], whilst we [15,19,20] and others [21] have interrogated EV cargo using sequencing and metabolomics [22]. There are few studies on the EV protein cargo in synovial fluid-derived EVs in OA [23,24] and none in the horse. We formerly identified that differential expression of small nucleolar RNAs (snoRNAs) [25,26,27] and microRNAs [28,29] contributes to this imbalance, which isa key mechanism in OA. We require biomarkers to identify early OA before cartilage ECM is irreversibly degraded. Our group has identified small non-coding RNAs (sncRNAs) distinguishing early equine OA synovial fluid [30] as well as identifying synovial fluid proteins as potential biomarkers [31,32,33,34,35,36].

Our work has described the small non-coding RNA EV cargo in a temporal model of equine OA [19]. In this present study, we have identified the differential EV protein cargo in synovial fluid and plasma from the same samples using SWATH-MS (Sequential Window Acquisition of all Theoretical Mass Spectra). This enabled in-depth local joint insight, whilst also enabling future translation for potential blood-based biomarkers. We test the hypothesis that synovial fluid and plasma-derived EV protein cargo can be used to identify OA at an early stage before clinical signs and irreversible cartilage degradation. Additionally, by determining changes in protein cargo in a longitudinal manner and combining this data using a multi-omics approach with small non-coding RNA EV data, we may be able to further elucidate the pathogenesis of early OA. 

## 2. Results

### 2.1. Model Outcome

We have previously described the synovial membrane score and histological evaluation of the third metacarpal bone [19,37] at the end point of the model (day 70 post induction). Briefly, the end point synovial membrane scores were significantly increased for the OA joints compared to control joints (*p* < 0.05), including cellular infiltration, intimal hyperplasia, subintimal oedema and total final histological score. Histological evaluation of the third carpal bone cartilage demonstrated significantly increased scores for the OA joints compared to control joints (*p* < 0.05), including chondrocyte necrosis, cluster formation, focal cell loss and final score.

### 2.2. EV Characteristics

EV characteristics for plasma and synovial fluid-derived EVs at all time points have been reported previously [19]. Briefly, nanoparticle tracking analysis identified the mean and mode particle size (±s.d. and range) for control synovial fluid to be 219.8 (±49.7, 114.6–360.3) nm and 180.9 (±44.5, 91.6–319.7) nm, OA synovial fluid 221.8 (±40.2, 154.9–312.1) nm and 195.0 (±64.8, 121.7–407.9) nm and plasma 158.4 (±18.9, 125.0–203.7) nm and 123.3 (±22.3, 90.4–163.2) nm. For synovial fluid-derived EVs, no statistical difference in size, size distribution or concentration between the control and OA joints was identified. However, for plasma-derived EVs, a reduction in the mean EV size was identified on day 49 (147.3, ±11.2, 133.4–158.2) nm compared to day 0 (167.1, ±18.5, 144.3–183.0) nm (*p* = 0.01). Exoview analysis identified positive staining for CD63 and CD81 markers on both synovial fluid and plasma EVs.

### 2.3. Characterisation of the Equine Plasma and Synovial Fluid EV Proteome

The EV spectral libraries initially identified 444 and 2271 proteins in plasma and synovial fluid, respectively. At the individual sample analysis level, 259 proteins were identified in at least one sample from plasma-derived EVs and 1201 from synovial fluid-derived EVs [19]. 

We filtered the data by two criteria: each protein needed complete observations for all samples for at least one time point and <30% missing values. This was so we could make statistical inferences on the same data downstream in the analysis [38,39]. After filtering, there were 331 proteins for synovial fluid-derived EVs and 203 for plasma-derived EVs. The overlap of proteins shared between plasma and synovial fluid-derived EVs after filtering and quality control is shown in Figure 1A. The cargo of the synovial fluid EVs was found to be more heterogeneous than plasma EVs and varied considerably sample to sample (Appendix A). Among all the filtered proteins, 102 overlapped between the synovial fluid EVs and plasma EVs. 

We carried out over-representation analysis (ORA) on the filtered proteins for the plasma (Figure 1B) and synovial fluid (Figure 1C), to identify which gene ontology (GO) terms were enriched in the EVs compared to the whole equine proteome [40]. ORA analysis identified several terms that were functionally enriched in EV cargo for both biofluids compared to the whole proteome including ‘complement activation classical pathway’, ‘complement activation alternative pathway’ and ‘serine-type endopeptidase inhibitor activity’. These terms are all in keeping with the known role of EVs driving tissue inflammation and repair [15,41,42,43,44] and support their potential as biomarkers of OA progression. However, there were also terms unique to each biofluid. In plasma, we saw several terms relating to lipoproteins and proteasome were enriched. In synovial fluid, the term ‘leukocyte cell-cell’ adhesion was uniquely enriched. Overall, this step served to establish that the proteins we identified were in keeping with known EV biology but also demonstrated that different equine biofluids have their own particular proteomic signature.

### 2.4. A Multivariate Approach Identified a Separation by Day for Proteins from Plasma-Derived EVs

After characterising the respective cargos of plasma and synovial fluid-derived EVs, we then investigated whether these cargos could be used to identify markers of OA progression across time. Firstly, we used multi-level principal component analysis (mPCA) to explore the variance in our data and whether this variance corresponded to a biological effect. We used mPCA over classical PCA as this accounts for intraclass correlation within horses across repeated measurements. Following multi-level decomposition, we derived principal components that captured the variance in our data. In plasma EVs, PC1 and PC2 accounted for 28% and 14% of the variance, respectively, and allowed us to see some separation by days since induction across both PCs (Figure 2A). Pearson’s correlation demonstrated moderate but significant correlation (0.43*) between PC1 and days since induction. In the synovial fluid-derived EVs, PC1 and PC2 captured 30% and 16% of the variance, respectively; however, this variance did not appear to correspond to days since OA induction (Figure 2B) or disease (Figure 2C), as there was no significant correlation between any of the PCs and biological factors. 

### 2.5. Differential Expression of Plasma and Synovial Fluid-Derived EVs

Next, we carried out differential expression analysis on the filtered proteomics data for plasma- and synovial fluid-derived EVs. ProDA was used for differential expression analysis as it is specifically designed to handle proteomics data with high numbers of missing values such as in our data [45]. It does this by modelling the missing values with a probabilistic drop-out model rather than directly imputing them, reducing the odds of false-positives. Table 1 shows the number of differentially expressed proteins in plasma and synovial fluid-derived EVs for each time point compared to day 0 (*p* < 0.05) (modelled as a series of pair-wise comparisons) as well as over the entire time course (time modelled as a continuous variable). Appendix A contains the identities of each protein. 

However, among these proteins Coagulation factor XIII A chain was the only significant protein after correcting for FDR (set at 0.05) in plasma-derived EVs at day 35. As only one protein was significant after correcting for FDR, we did not carry out any functional analysis on the differential expression results. This study was low-powered, limiting the amount of statistical inference we could do on the proteomics data, so instead we opted for multi-omics integration to see if we could capture any latent factors that correspond to OA progression or any features that co-vary across OA progression. 

### 2.6. Small RNA Sequencing Analysis Quality Control Prior to Multi-Omics Analysis

To carry out multi-omics integration, we used our previously published small RNA sequencing data, as these data are matched and come from the same samples (a requirement for many multi-omics techniques). Prior to using the small RNA sequencing data in the multi-omics integration, we carried out a quality control step by diagnosing the source of missing values (which was found to correspond to sequencing depth) and then filtering again by missingness (Appendix A). 

We next carried out PCA to see if our remaining sequencing data captured relevant biological variance. Following filtering, there were only two remaining plasma-derived EV snoRNAs, snoRNA U3 and snord58, and two synovial fluid-derived EV snoRNAs, snoRNA U3 and snord58 [14]. Therefore, we did not carry out PCA for the snoRNAs. The clustering of samples in PCA space demonstrated no separation by days since induction, disease, or horse for miRNA (Figure 3A), lncRNA (Figure 3B) or snRNA (Figure 3C). When all sequencing data were combined (Figure 3D), plasma-derived EV samples showed a mild separation by time across PC1 (24% variance explained). Furthermore, synovial fluid-EVs appeared to show time separation, but this did not appear to be OA-specific. Overall, the PCA results suggested that there was a mild biological signal in the data that could be used for multi-omics integration.

### 2.7. Multi-Omics Time Series Reveals Patterns of Expression during OA Progression

Multi-omics time series clustering was used to identify which features (proteins and small RNAs) co-vary together over time, and whether any of these clusters appear to correspond to OA progression. This allowed us to identify potential global trends in our data.

Multi-omic time series clustering was undertaken for plasma and synovial fluid-derived EVs using the mfuzz algorithm (Figure 4). The ideal number of clusters was identified using the elbow method. For plasma-derived EVs, cluster 4 appeared to represent a time-dependent response to OA induction after day 10. The features with the highest membership for this cluster were proteins. Clusters 2 and 3 had very similar profiles, showing low variance across time, and contained a higher proportion of small non-coding RNAs. Cluster 1 had a distinct profile, though as it fluctuated it was difficult to speculate how it translated to OA progression (Figure 4A,B). For the synovial fluid-derived EVs, there were two clusters corresponding to an initial response at OA induction at day 10 (clusters 1 increased at day 10 and cluster 3 decreased at day 10). It appeared that clusters 5 and 6 were redundant due to a correlation between their centroids, whilst clusters 4 and 5 had inverse profiles at day 10 (Figure 4C,D).

To validate whether any of these clusters corresponded to a biological response rather than noise, we carried out functional enrichment on their protein members. For plasma-derived EVs, only cluster 4 had any significant GO terms associated with it.These were primarily associated with organelles and supramolecular complexes (Figure 5A). For synovial fluid-EVs, only clusters 1 and 3 had significantly enriched GO terms. Cluster 1 was highest at 10 days and appeared to be an immune response to the initial intervention (Figure 5B). This is reflected in the enriched terms (immune response, humoral immune response, complement activation). 

### 2.8. Multi-Omics Factor Analysis Demonstrates That Proteins Drive the Bulk of EV Cargo Variation during OA Progression

Multi-omics time series clustering revealed that there were features (small RNAs and proteins) that co-varied together across time after OA induction and that some of these features appeared to correspond to a biological response. Next, we employed multi-omics factor analysis for plasma and synovial fluid-derived EVs to see if we could reduce the data to latent factors that captured the variance associated with OA progression. Latent factors represent new features that are combinations of the original features from the data (protein and small RNA) and can capture information that would otherwise be hidden. The factors are composed of feature weights and can be used to extract knowledge from complex datasets. In this case, a factor might correspond to the variance associated with OA progression. We can then inspect the feature weights to see if any proteins/RNAs drive the bulk of the variance. 

We used MOFA2 to carry out the multi-omics factor analysis [46]. The MOFA2 model produced six factors for both datasets, which were sufficient to capture the variation present. For plasma-derived EVs, the proteomics data explained 50% of the variance and the sequencing data explained 15% of the variance (Figure 6A). Factor 1 and factor 3 were found to be somewhat correlated with each other. 

For synovial fluid-derived EVs, we employed a multi-group structure to capture sources of variation within the control and OA groups over time. Interestingly, in the control group, the small non-coding RNA sequencing data contributed to a higher proportion of the variance (>10%) than in the OA group (<5%) (Figure 6D–F). This suggested that changes in protein cargo might be a more important driver of OA progression than changes in the small non-coding RNA cargo. 

The factor data for plasma-derived EVs protein heatmap for factor 1 are shown in Figure 7A,B, which shows the Pearson correlation between covariates in the phenotype data and latent factors. For both plasma and synovial fluid-derived EVs, the proteomics and small non-coding sequencing data showed little integration for the latent factors (Figure 7C and Figure 8C). Generally, the variance captured by each factor was explained by either the proteomics data or the small non-coding sequencing data, rather than a combination of the two. This was demonstrated by the weightings for factor 1, where the highest weighted proteins had a weighting > 0.8 and the highest weighted small non-coding RNAs had a weighting < 0.025. This suggested the two-omics had different modalities and were not correlated with each other. This complemented findings from the multi-omics clustering analysis.

We correlated the latent factors with phenotype (day, disease status) data. For the plasma-derived EVs, factor 1 was significantly correlated with day and factor 3 slightly correlated with day (in agreement with the two factors being correlated with each other). Factors 5 and 6 were correlated with horse (Figure 7B). It appeared most of the variance that correlated with phenotype came from the proteomics data (Figure 7C). The weightings for the proteins contributing to factor 1 can be seen in Figure 7D. Figure 7E,F shows the loadings of the top features of factors 2 and 1 in sequencing data, respectively. 

For synovial fluid-EVs, factor 2 captured the variance between horses and was mainly due to protein cargo (Figure 8A). For the synovial fluid-derived EVs, factor 6 was most strongly correlated with day (Figure 8B), with factors 3 and 4 also being weakly correlated with day. For factors 6 and 4, most of the variation came from the sequencing data and for factor 3, most of the variation came from the proteomics data. The separation by day for factor 3 was not very pronounced; however, there did appear to be a tendency for the abundance of these proteins to be lower in the OA samples (Figure 8C,D). For factor 6, the separation by day was also not especially pronounced; however, there appeared to be a trend for the abundance of these proteins to be lower in the OA samples (Figure 8D). 

The weights of the different features contributing to factor 6 for sequencing are shown in Figure 8E and contributing to factor 3 for protein data in Figure 8F. 

As we established that the latent factors captured variation associated with features in the phenotype data, especially for the plasma-derived EVs, we next wanted to map this to biological processes to understand how the latent factors might drive OA progression. We first used Cluster Profiler’s compare Cluster function to carry out simultaneous GSEA on all the proteins contributing to the latent factors for plasma- and SF-derived EVs. Only factor 1 for plasma-derived EVs and factors 2 and 3 for SF-derived EVs had significantly enriched GO terms, perhaps reflecting that these factors captured relevant biological variation rather than technical variation. Notably, plasma-EV factor 1 showed significant enrichment for lipid metabolism (Figure 9A), indicating that EVs may modulate lipid metabolism in OA. SF-EVs factor 2 only had one enriched GO term, protein-containing complex assembly (GO:0065003), but this did not appear to correlate with any phenotype data. SF-EVs factor 3 showed enrichment for intermediate filaments and supramolecular complexes (Figure 9C). As this factor showed slight correlation with day, this could reflect tissue repair in response to induction of OA or normal EV cargo transport during tissue turnover. Factor 6 did not have any enriched GO terms, most likely because small ncRNAs derive the variation in this factor. 

### 2.9. Synovial Fluid-Derived EVs Induced Intracellular Chondrocyte Signaling

To validate whether EVs from the model have functional effects, synovial fluid-derived EVs from the model were analyzed for the activity of 13 transcription factor complexes representative of specific signalling pathways related to OA in SW1353 cells [47]. 

Four reporters showed potential changes with OA induction; CRE, NFKB, SRE, SRF (Figure 10A–D). Next, we fit a linear fit interaction model to determine whether there was an interaction between reporter value and control or OA-derived synovial fluid EVs over time. All four reports displayed a time dependent increase in reporter activity in OA-derived synovial fluid EVs. Figure 10F–I shows the fitted model for CRE, NFKB, SRE and SRF. There was a significant interaction between time and OA for each reporter shown following FDR correction (Figure 10J). 

## 3. Discussion

This is the first study to our knowledge that has interrogated temporally extracellular vesicles in the development of OA in any species. Our combined analysis of omics datasets has increased the understanding of the role of EVs in OA. We implemented a multi-omics approach, integrating the proteomics data with our previously published small non-coding RNA sequencing data to capture underlying trends occurring in OA EVs during disease progression. Multi-omics time series clustering and factorisation captured variance that was associated with time since OA induction. One of the primary findings of our research approach was the demonstration that EV protein cargo appears to be more important during OA progression than small non-coding RNA cargo. There was no correlation between the two. Our novel synovial fluid versus plasma approach makes it possible to identify plasma measurable biomarkers that reflect the synovial fluid status. We undertook a DIA proteomics approach for our EVs. Using this, we identified 2271 and 444 proteins in the synovial fluid and plasma libraries, respectively. These are similar numbers to other studies [48,49]. Thus, we identified significantly fewer proteins in plasma-EVs. Proteomic characterization of plasma-derived EVs is challenging due to the presence of a few highly abundant plasma proteins, limiting the detection of lower abundant proteins. Indeed, 85% of the mass spectrometric data corresponds to the seven most abundant proteins together [50]. Others have demonstrated that plasma depletion prior to EV isolation provides slightly elevated protein coverage (474 vs. 386) [49], an approach that could be useful for future studies. 

The spectral data were characterised by a large number of missing values, especially for the synovial fluid-derived EVs. Therefore, in our proteomics analysis we attempted to determine whether the missing values were due to sampling error, technical limitations or a biological factor by replacing the missing values with zeroes and carrying out unsupervised clustering. We assumed that if the missing values clustered by a biological effect, they were biological zeroes and could be included in downstream statistical analysis. However, the clustering appeared to be at random and the missing values were most likely due to sampling error (Appendix A).

It was notable that whilst fewer proteins were identified in the plasma-derived EVs, overall these proteins were identified more consistently in all of the samples (Appendix A). For synovial fluid-derived EVs, the maximum number of proteins consistently identified in the samples was ~50 (Appendix A). This could be reflective of the technical difficulty of working with synovial fluid compared to the established biofluid plasma, or due to the increased abundance of EVs in plasma compared to synovial fluid [19]. However, there is also some evidence that sampling at a disease site provides a more complex mixture of EVs (and thus perhaps cargo) than sampling from a circulating source [51]. Indeed, interrogation of the plasma and synovial fluid protein libraries identified surface marker proteins for joint cells. We hypothesised that the source of the EVs in the synovial fluid (multiple cell types adjacent to the fluid; chondrocytes, synoviocytes, mesenchymal stem cells, bone) could contribute more to variability than the EV cargo. Synovial fluid-derived EVs contained more surface markers for chondrocyte, synoviocyte, mesenchymal stem cells than plasma, and synovial fluid-EVs alone contained bone surface markers (Appendix A).

Interestingly, despite the differences in plasma and synovial fluid-derived EV protein cargo, many of the proteins mapped to the same GO terms, including proteins relating to serine endopeptidase activity, which we had previously shown were modulated by mesenchymal stem cell treatment [22]. Among the plasma-derived EVs, GO terms related to lipid homeostasis (low-density lipoprotein particle, very-low-density lipoprotein particle, spherical high-density lipoprotein) were more enriched compared to synovial fluid-derived EVs. Apart from our group, there is only a single study which interrogated globally EV cargo in equine OA. The group found that in EV-enriched pellets, saturated fatty acids palmitic acid, stearic acid and behenic acid indicated OA when comparing samples from horses with clinical disease. These observed fatty acid modifications could be detrimental and contribute to inflammatory processes and cartilage degradation in the disease [52]. Utilising the same samples using a targeted approach, they also identified an inverse relationship between the OA grade and hylauronic acid–EV count [53]. Coagulation factor XIII A chain was the only significant protein differentially expressed after correcting for FDR. This was in plasma-derived EVs at day 35. This may be due to the small number of horses used in the large animal study and the nature of this temporal study. Coagulation factor XIII A chain promotes cross-linking between fibrin molecules at the final stage of the blood coagulation cascade. Macrophages in the synovium are one of the sources of Coagulation factor XIII A [54]. Interestingly, the plasma concentration levels of this protein were associated with inflammatory arthritis and cartilage breakdown [55]. When comparing our data to the literature, few studies have interrogated in an unbiased manner the EV proteome of synovial fluid-EVs and none temporally. However, one human study did identify haptoglobin, orosomucoid and ceruloplasmin as significantly up-regulated, and apolipoprotein down-regulated in OA-EVs derived from female synovial fluid. In males, they identified β-2-glycoprotein, and complement component 5 proteins as up-regulated and Spt-Ada-Gcn5 acetyltransferase-associated factor 29 as down-regulated in male OA EVs. However, this study did not apply FDR correction to their findings and the EVs were isolated using ultracentrifugation alone, which is known to increase contamination issues [56]. In our study, the horses consisted of two gelded males and two mares; therefore, inference of sex could not be determined. 

One of the advantages of our methods was that by reducing the data to latent factors, we identified features in the data that may have been otherwise hidden. This is the first study to describe that during OA progression, plasma EV content has altered lipid homeostasis, and this was despite no change in EV numbers [19]. Lipid metabolism is closely related to the progression of OA [57], but the role of EVs in this process remains unclear.

This study has a number of limitations. As a large animal model, the study was limited in terms of animal numbers. This was compounded by the longitudinal nature of the data so that few proteins were significantly differentially abundant following FDR correction. The volume of biofluids collected was limited and had been used in our previous work. This, together with its use in multiple experiments, meant there was none left to validate our findings with other methodologies. 

When combining small RNA sequencing and proteomics data, there appeared to be mild time series patterns of expression during OA progression. Following OA induction for plasma-derived EVs, there was an initial decrease at 10 days followed by an increase to day 63 in cluster 4. This was largely composed of proteins associated with organelles and supramolecular complexes. The assembly of supramolecular complexes is associated with signal transduction and regulates many signalling events (reviewed [58]). It could be that the molecules within EVs during OA progression in our model are contributing to the pathways regulating inflammation and cell death/survival. This is because it is known that the regulation of the assembly and silencing of these complexes plays important roles in these events. For the synovial fluid-derived EVs, there were two clusters which correlated to an initial response at day 10. However, only cluster 3 had significant GO-terms associated with it. These related to changes in immune response (humoral immune response, complement activation); we predict these were due to the initial intervention. Day 10 was the first time point taken, so the changes that were occurring in EV content before this are unknown and warrant further study. 

The use of luciferase reporter assays for studying cellular responses in biofluids such as synovial fluid has been presented in relatively few studies [59]. This is the first time to our knowledge that a luciferase reporter has been used to determine the biological responses in EVs. We aimed to determine if signaling pathways could be affected directly by EVs from our OA model. These assays could inform on relevant signaling pathway activity promoted by extracellular vesicle cargo. The use of these assays could give information on signaling pathways. We had previously used the assay to undertake the first mechanistic comparison between non-OA and OA synovial fluid. Here, we highlighted that MAPKs, cPKC/NFkB and PI3K/AKT were crucial OA-associated intracellular signaling routes [47]. There was a significant interaction between time and disease status of the synovial fluid-EVs for cAMP response element (CRE), Nuclear factor-kappa B response element (NFKB), serum response element (SRE) and serum response factor response element (SRF) reporters. As time progressed, CRE, NFKB, SRE, SRF reporter activity increased. When analyzing the static time points, this appeared to be the greatest at day 56 but was not significantly different. This may have been to the small number of donors for each time point or the large data variation. As the changes particularly at day 56 looked convincing, we further interrogated the data using a linear model and identified a significant interaction between OA-derived EVs and time. There was a significant interaction between time and OA for CRE, NFkB, SRE, SRF. Thus, transcription factor activity analysis identified increased activation of NFkB, MAPK (i.e., SRE, CRE), RhoGTPase (i.e., SRF) and PKA (i.e., CRE) signaling in EVs derived from synovial fluid in a time-dependent manner. Interestingly, in our previous study, we described that NFkB, SRF and SRE were more strongly induced by OA-synovial fluid than non-OA-synovial fluid. CRE and SRE are often associated with cellular growth and differentiation. CRE is responsive to the transcription factors ATF and CREB, which can be activated by both MAPK signaling and cAMP/PKA signaling [60]. Th While SRE activation requires both MAPK/ERK and RhoA/ROCK signaling, SRF-RE is responsive to RhoA/ROCK alone [61]. We propose, based on our previous studies, that equine synovial fluid-EVs could induce phenotypic processes in chondrocytes including chondrocyte dedifferentiation predominantly driven by ERK signaling via SRE but also via PI_3_K, ROCK and JNK via activation of SRF and CRE. Chondrocyte fibrosis may be affected, directed via JNK and CPC driven by CRE and NFkB, respectively. Inflammatory responses characterised by transcription of chemokine genes dependent upon cPKC signalling through activation of SRE and NFkB as well as ERK activationvia SRE could be affected. Finally, extracellular matrix degradation mediated via cPKC and ERK signalling through activation of NFkB and SRE could be downstream consequences. However, it is not proven to what extent the EVs within SF contribute to the signaling induction evident here. We propose to study the ability of extracellular vesicles from a variety of sources to effect the chondrocyte and synoviocyte phenotype. It is clear that EV cargo comprising integrated sets of biological information can impact cell signaling. As we have demonstrated that OA-synovial fluid can induce chondrocyte proliferation and protein translation in human articular chondrocytes (HACs) to a greater degree (double) than non-OA-synovial fluid [47], it would be beneficial to determine if EVs from these sources elicit the same effects. 

## 4. Materials and Methods

All reagents are Thermo Scientific, Altringham, UK unless otherwise stated.

### 4.1. Horses and Study Design, Induction of Osteoarthritis

All experiments are fully documented in our previous manuscript [19]. In brief, Danish Animal Experiments Inspectorate (permit 2017-15-0201-01314) and the local ethical committee of the Large Animal Teaching Hospital, University of Copenhagen approved the experimental protocol. Procedures were undertaken according to EU Directive 2010/63/EU for animal experiments. 

Four skeletally mature Standardbred trotters were included in this study (two mares and gelded males). As previously described, OA was surgically induced in the left middle carpal joint and the right middle carpal joint underwent sham surgery [62]. Plasma plus synovial fluid, sampled from both middle carpals before and following OA induction were collected. Two weeks following OA induction, horses were exercised with 2 min of trot (4.4–5.3 m/s), 2 min of fast trot/gallop (9 m/s) and 2 min of trot (4.4–5.3 m/s) for 5 days/week on a treadmill.

### 4.2. Synovial Fluid and Plasma Sampling

Synovial fluid and plasma samples were obtained from both middle carpal joints prior to (day 0) and 10, 35, 42, 49, 56, 63 days following surgery. Biofluids were processed within one hour and stored at −80 °C immediately following centrifugation.

### 4.3. Post-Mortem Examination

Following euthanasia, cartilage and synovium from the middle carpal joints were placed in neutral buffered 10% formalin and processed for histology with staining for haematoxylin and eosin and safranin O (cartilage only). Grading of the synovial membrane and cartilage was performed [2]. 

### 4.4. EV Isolation

The synovial fluid and plasma collected on days 0, 10, 35, 42, 49, 56 and 63 from OA and control joints were thawed, and synovial fluid was subsequently treated with 1 μg/mL hyaluronidase (from bovine testes; Sigma–Aldrich, Gillingham, WI, USA) for 1 h at 37 °C. Both the synovial fluid and the plasma were centrifuged at 2500× *g* for 10 min and 10,000× *g* for 10 min at 4 °C. EVs were subsequently isolated by size exclusion chromatography using qEV single columns (IZON, Lyon, France) following the manufacturer’s instructions. Briefly, 3.5 mL of phosphate-buffered saline (PBS; Sigma–Aldrich, Gillingham, United Kingdom), previously processed using a 0.22-μm polyethersulfone filter (Sartorius, Göttingen, Germany) was passed through the column, followed by 150 μL of synovial fluid or plasma. The first five 200 μL flowed through fractions were discarded, and the following five 200 μL fractions pooled (isolated EVs). Isolated EVs were subsequently concentrated to a volume of 100 μL using a 2 mL 100,000 kDa Vivaspin column (Sartorius).

### 4.5. EV Characterisation

EVs were characterised using nanoparticle tracking analysis (Nanosight NS300, Malvern Panalytical Ltd., Malvern, UK) and Exoview (Nanoview, Malvern, UK), as described previously [19].

### 4.6. Plasma and Synovial Fluid-Derived EV Spectral Library Preparation for Proteomics

EVs were isolated from an 8 mL pool of all study plasma samples. For synovial fluid-derived EVs, 11 mL equine synovial fluid was used. This was a SF pool derived from our equine musculoskeletal biobank (VREC561), with samples collected from our Clarke et al. study [15] and samples from the present study. For both libraries, the pools were centrifuged at 100,000× *g* and 4 °C for 70 min and then resuspended in 200 µL 6M urea/1M ammonium bicarbonate/0.5% sodium deoxycholate. The sample was transferred to a 1.5ml Eppendorf and sonicated to disrupt the vesicle membranes. An aliquot of 10 µL was analysed by Sodium dodecyl-sulfate polyacrylamide gel electrophoresis (SDS-PAGE), resulting in a total protein estimate of 100 µg. The sample was reduced and iodoacetylated prior to the addition of 4 µg Trypsin/LysC (Promega < Southampton, UK). It was incubated at 37 °C for 3 h, urea concentration was reduced by the addition of 1 mL of water, and then incubation continued overnight at 37 °C. The next day, 4 mL cation exchange buffer A (10 mM KH_2_PO_4_/25% ACN pH3) was added and the pH was adjusted to 3 with concentrated phosphoric acid. The sample was filtered through a 0.2 µm filter and the entire sample was loaded onto a Polysulfoethyl. A strong cation-exchange column (200 × 4.6 mm, 5 µm, 300 Å; Poly LC, Columbia, SC, USA) was installed on an Agilent 1100 system (Agilent Technologies, Santa Clara, CA, USA). Peptides were eluted using a gradient from 0–15% solvent B (1 M KCl in 10 mM KH_2_PO_4_/25% ACN pH3) in 45 min and 15–50% solvent B in 15 min at a flow rate of 1 mL/min, and 2 mL fractions were collected. The fractions were evaporated to dryness and desalted using a macroporous C_18_ high-recovery reversed phase column (4.6 × 50 mm, Agilent Technologies, Santa Clara, CA, USA) installed on an Agilent 1260 system before being dried once more. The peptides were resuspended in 10 µL 0.1% formic acid just prior to liquid chromatography tandem mass spectrometry (LC-MS/MS) analysis.

Half of each of the 16 fractions was loaded onto a trap column (nanoACQUITY UPLC Symmetry C_18_ Trap Column (Waters, Manchester, UK)) installed on an Eksigent NanoLC 415 (Sciex, Framingham, MA, USA) and washed for 10 min with 2% ACN/0.1% formic acid. Peptides were separated on a bioZEN 2.6 µm Peptide XB-C_18_ (FS) nano column (250 mm × 75 µm, Phenomenex, Torrance, CA, USA) using a gradient from 2−50% ACN/0.1% formic acid (*v*/*v*) over 120 min at a flow rate of 300 nL/min. Data-dependent acquisition was performed on a Triple TOF 6600 (Sciex, Frammingham, MA, USA) in positive ion mode using 25 MS/MS per cycle (2.8 s cycle time), and the data were searched using ProteinPilot 5.0 (Sciex, Frammingham, MA, USA) and the Paragon algorithm (Sciex, Frammingham, MA, USA) against the horse proteome (UniProt Equus cabullus reference proteome, 9796, May 2021, 20,865 entries). Carbamidomethyl was set as a fixed modification of cysteine residues and biological modifications were allowed. The data were also searched against a reversed decoy database and proteins lying within a 1% or 5% global false discovery rate (FDR) were included in the library. For retention time alignment of SWATH data, the library was regenerated with no biological modifications allowed.

### 4.7. Plasma and Synovial Fluid-Derived EV Sample Preparation for SWATH-MS

EVs isolated from individual plasma and synovial fluid samples were centrifuged at 100,000× *g* and 4 °C for 45 min and were then resuspended in 100 µL 6 M urea/1 M ammonium bicarbonate/0.5% sodium deoxycholate. The samples were sonicated and 10 µL was set aside for analysis by SDS-PAGE. Following reduction and iodoacetylation, samples were incubated with a 1:1 mixture of hydrophilic and hydrophobic magnetic carboxylate SpeedBeads (Cytiva, Marlborough, MA, USA) (12 µL beads in total) in the presence of 50% ethanol for 1 h at 24 °C and 1000 rpm. After washing with 80% ethanol, the beads were resuspended in 100 µL 100 mM ammonium bicarbonate and 2.4 µg trypsin/LysC was added. The samples were incubated overnight at 37 °C and 1000 rpm. The supernatants were collected, acidified by the addition of 900 µL 0.1% trifluoroacetic acid and desalted as above. The dried fractions were resuspended in 10 µL 0.1% formic acid just prior to LC-SWATH-MS analysis.

Aliquots of 5 μL were delivered into a TripleTOF 6600 mass spectrometer (Sciex, Frammingham, MA, USA), as described above, and the same 2 h gradient was applied. SWATH acquisitions were performed using 100 windows of variable effective isolation width to cover a precursor *m*/*z* range of 400–1500 and a product ion *m*/*z* range of 100–1650. Scan times were 50 ms for TOF-MS and 33 ms for each SWATH window, giving a total cycle time of 3.7 s. Retention time alignment and peptide/protein quantification were performed by Data-Independent Acquisition by Neural Networks (DIA-NN) 1.8 [63], using the same reference horse proteome as above to reannotate the library, a precursor FDR of 1% and with a match between runs and unrelated runs selected. Proteins with missing values were removed from the protein group matrix prior to analysis using Partek Genomics Suite 7.18 (Partek Inc, Chesterfield, MO, USA).

Proteomics data are available via ProteomeXchange with identifier PXD041515.

### 4.8. Small RNA Sequencing of EVs

Plasma and synovial fluid-derived EV small non-coding alterations in our temporal model of OA using the same samples analysed with proteomics have been previously published [19] and are also available at GEO ID GSE200330.

We investigated whether the missing values evident in the small RNA-sequencing data (microRNAs, snRNAs, snoRNAs, lncRNA) were true zeroes or sampling zeroes (due to sequencing depth). Pearson correlation of the total number of missing values by sequencing depth and principle component analysis (PCA) with complete observations vs PCA with missing values as zeroes were used. Both suggested the zeroes were sampling zeros rather than biological zeros (Appendix A). Therefore, for further data processing, we filtered rows that had >60% zeros and replaced the zeros with NAs. The threshold for missingness was set at this level due to the lower number of features and MOFA2′s robustness to missing values. As we only had tRNA data available for the plasma, we did not include this class of small RNAs in further analysis.

### 4.9. Reporter Gene Assay

To determine the functional effects induced by extracellular vesicles derived from synovial fluid, we analysed the activity of transcription factor complexes representative of specific signalling pathways we have previously demonstrated to have a role in OA [47,64]. Stably transduced response element (i.e., NFκB-RE, SBE, NFAT5-RE, TCFLEF-RE, CRE, ARE, AP1, SRE, SRF-RE, SIE, ISRE, GRE, NB) driven Nano luciferase SW1353 (HTB-94, ATCC) reporter cell lines were generated according to a previously published protocol [47]. Reporter cells were trypsinized and re-seeded (60.000 cells/cm^2^) into 384-well plates (Greiner Bio-One, Stonehouse, UK) and cultured overnight with Dulbecco’s Modified Eagles Medium (DMEM/F12) supplemented with 0.5% foetal calf serum. Serum-starved cells were stimulated with 10% synovial fluid-derived extracellular vesicles for six hours. Samples used for this study were identical to those used for proteomics; synovial fluid was obtained from both middle carpal joints prior to (day 0) and 10, 35, 42, 49, 56, 63 days following surgery (n = 4 donors). Stimulation with 0.1% foetal bovine serum was used as a control. After stimulation, cells were lysed using 15 µL Milli-Q. Following the addition of Nano-Glo reagent (1:1 ratio; Promega, Southampton, UK), luminescence was quantified using the Tristar^2^ LB942 multi-mode plate reader (Berthold Technologies, Bad Wildbad, Germany).

### 4.10. Statistical Analysis

#### 4.10.1. Single-Modality Approaches

All statistical analysis was carried out using the R statistical programming environment [15]. During quality control of the proteomics data, different normalisation methods were assessed using NormalyzerDE [65]. Due to the high number of missing values in some samples shifting the distributions higher (assessed using the DEP package), data were log2 transformed [66]. We assessed whether missing values were absent completely at random (MCAR), missing not at random (MNAR) or missing at random (MAR) by replacing them with zeroes and carrying out unsupervised clustering. The samples were then filtered in a group-aware manner using the DEP package so that each protein needed complete observations for at least one time point, followed by an additional filtering step to remove any proteins with >30% missing values.

Differential expression analysis was carried out using the proDA package [36]. We used proDA in favour of other more established packages due to the sparsity of the data, as proDA implements a powerful probabilistic drop-out model to handle the missing values. We checked for agreement in our results using the more established package: limma. For both packages, we modelled time as a continuous variable and as a series of pair-wise comparisons to account for linear and non-linear responses. All statistical tests were corrected for FDR using the Benjamini–Hochberg method. Results were considered significant at 5% FDR.

#### 4.10.2. Statistical Analysis of Reporter Data

Reporter intensities were expressed as ratios to intensities observed following stimulation with 0.1% foetal bovine serum. To ascertain whether a reporter had biologically relevant levels of expression, normalised ratios were ordered and plotted from highest to lowest. NFkB, SRE, SRF and CRE reporters were retained for downstream analysis. Univariate statistical analysis for each reporter was performed using a Wilcoxon rank test to assess differences between Control and OA samples at each time point. Noting a time-dependent effect, further linear interaction models were built to assess the interaction between Control and OA samples for each reporter with time as a continuous variable.

#### 4.10.3. Multi-Omics Time Series Clustering

For both stages of the multi-omics analysis, we integrated the proteomics data obtained here with our previously published sequencing data. The processed count data was TMM-normalised to account for sequencing depth, then converted into log-CPM values using the EdgeR-limma workflow [67].

To carry out time series clustering, the proteomics data were mean and variance filtered to remove proteins that had low expression and low variance (as clustering methods are sensitive to noise). The proteomics and sequencing data were then Z-score transformed and concatenated following previously published guidance [68]. The combined data were analysed using the mfuzz soft-clustering algorithm and cross-validated using the dtwclust hard-clustering algorithm [69]. The number of clusters were selected using the elbow method (four or less for plasma and six or less for synovial fluid).

#### 4.10.4. Multi-Omics Factor Analysis

Multi-Omics Factor Analysis (MOFA) model training was undertaken on the 26 plasma samples and 52 synovial fluid samples using the MOFA2 package [46]. The inputs for the MOFA model training comprised 99 non-coding RNAs, 203 proteins for plasma and 117 non-coding RNAs and 331 proteins for synovial fluid. Package default options were used for training except for convergence mode, which was set to “slow” and max iterations, which were set to “3000”. Due to the small sample size, we set the number of factors to six. Each latent factor had to explain a minimum of 2% variance in at least one data time. As the synovial fluid design was more complex, with treatment and control across time, we used the MEFISTO implementation of MOFA2 with a multi-group design [70]. Latent factors and feature loadings were then extracted for down-stream analysis.

#### 4.10.5. Functional Enrichment

Over-Representation Analysis (ORA) and Gene Set Enrichment Analysis (GSEA) were carried out using the ClusterProfiler package [71]. For the initial protein identification analysis, we used the entire equine proteome as the background. For later analysis, we used the filtered protein list from the relevant biofluid as the background. For the GSEA, the proteins were ranked by fold-change or factor loading for the differential expression and factor analysis, respectively. Gene ontology (GO) term annotation for the proteins was extracted from the Uniprot database. The full parent-child term map was then built using the *buildGOmap* function. We used the pRoloc package to convert GO identifications to names.

#### 4.10.6. Visualisation

For PCA and multi-level PCA, we only used data that had complete observations. Any rows with missing values were filtered out. The plots that depicted study data were created using ggplot, pheatmap or the VennDiagram packages in R.

## 5. Conclusions

Ultimately, EVs have numerous potential roles in OA clinical research and application, including the identification of disease specific biomarkers, the administration of non-diseased EVs as a targeted therapeutic, the blocking of EV signaling cascades and EV release, the disruption of EV cargo using novel therapeutics to prevent disease progression and as a surrogate for drug delivery. This study has identified a series of proteins which are differentially abundant between control and diseased plasma- and synovial fluid-derived EVs. Thus, these proteins may not only have potential application as markers of OA status and progression, but may ultimately help to further characterise the differences between healthy and OA EVs, aiding the compositional development of synthetic non-diseased EVs as a therapeutic. Crucially, this study has also identified that OA synovial fluid-derived EVs exhibit a differential effect on four response elements, which upstream are involved in G protein-coupled receptor, cytokine and growth factor signaling. This therefore demonstrates potential mechanistic disruptions to normal chondrocyte signaling caused by OA synovial fluid-derived EVs within the joint. Thus, this evidence supports the hypothesis that therapeutic targeting of OA synovial fluid-derived EVs or their cargo may provide a viable option for disease-specific OA treatment.

## Figures and Tables

**Figure 1 ijms-24-14888-f001:**
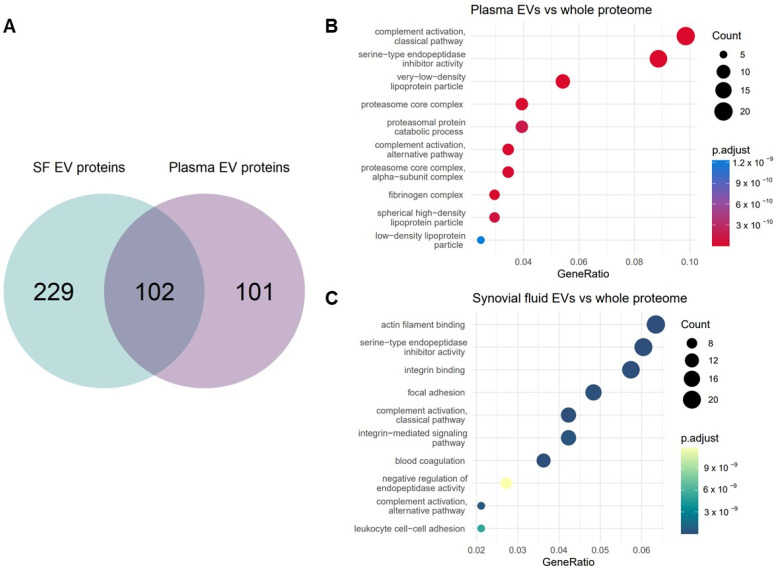
Proteins identified in plasma and synovial fluid-derived EVs. (**A**) Venn diagram showing shared and discrete proteins identified in EVs from each biofluid after filtering. Proteins were first filtered using the package DEP and then by filtering by % missing values (rows with >30% missing values were filtered). (**B**) Dotplot of ORA results from plasma-derived EV proteins versus the whole equine proteome. (**C**) Dotplot of ORA results from synovial fluid-derived EV proteins versus the whole equine proteome. The colour of dots relates to the adjusted *p*-value, size of dots relates to the number of proteins that map to each GO term.

**Figure 2 ijms-24-14888-f002:**
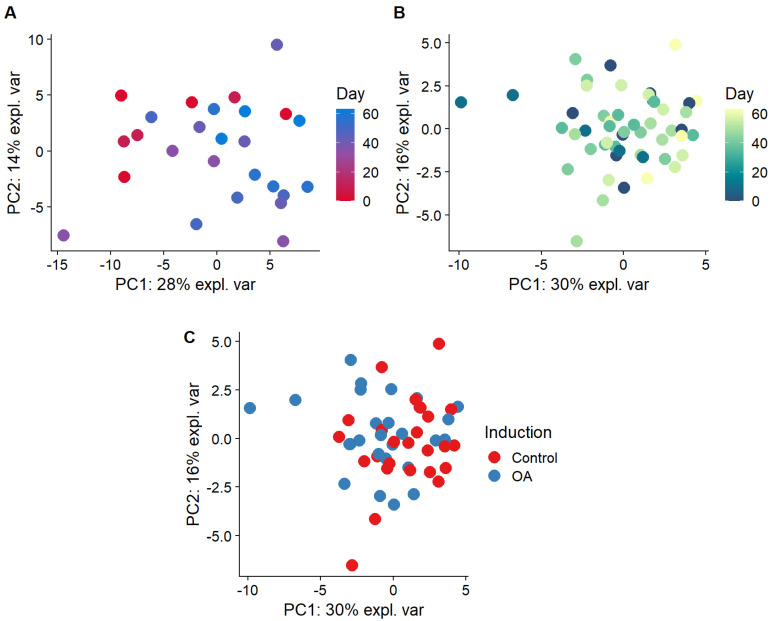
Scatterplots showing the mPCA results of plasma and synovial fluid-derived EV proteomics data. The x-axis corresponds to the variance captured by PC1 and the y-axis corresponds to the variance captured by PC2. The samples are projected into PC space and coloured by biological factors to allow us to visualise if the variance corresponds to features of biological interest from the phenotype table. (**A**) Plasma-derived EVs by time, (**B**) synovial fluid-derived EVs by time, (**C**) synovial fluid-derived EVs by disease. The first two principal components are plotted. Samples based on SWATH-MS. Each plotted point represents a sample, which is coloured by the time following OA induction (days) and disease status control and osteoarthritis (OA). Treatment relates to the number of days following OA induction.

**Figure 3 ijms-24-14888-f003:**
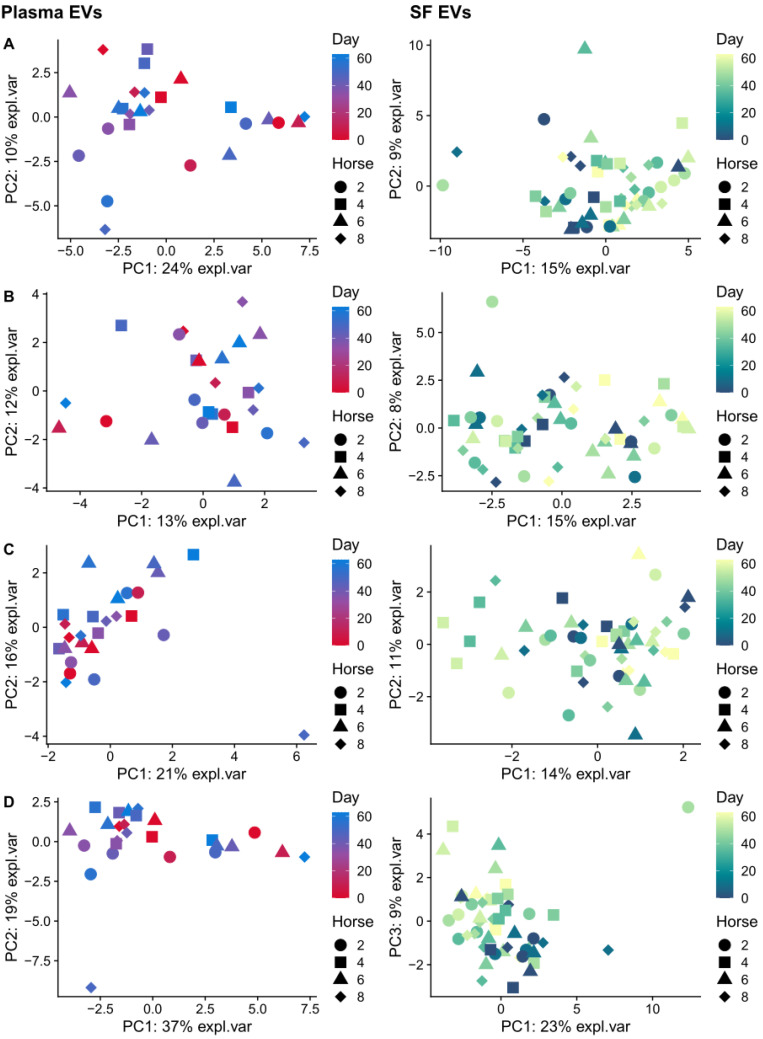
Exploratory data analysis using PCA plots. (**A**) microRNA, (**B**) lncRNA, (**C**) snRNA, (**D**) all sequencing data combined with plasma-derived and synovial fluid-derived EVs shown for each class. The first two principal components are plotted. The x-axis corresponds to the variance captured by PC1 and the y-axis corresponds to the variance captured by PC2. Samples based on small non-coding RNA sequencing data. Each plotted point represents a sample, the different horses are represented as shapes and coloured by the time. Red–blue gradient represents plasma samples, blue–yellow gradient represents synovial fluid samples.

**Figure 4 ijms-24-14888-f004:**
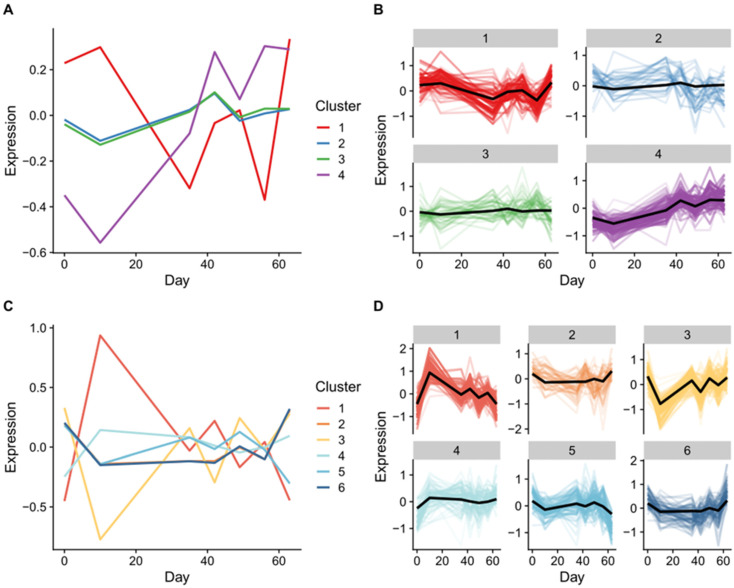
Time series multi-omics clustering using MFuzz. (**A**) Plasma centroid clustering using fuzzy c-means, (**B**) per cluster membership for plasma, (**C**) synovial fluid centroid clustering using fuzzy c-means, (**D**) per cluster membership for synovial fluid. For per cluster membership, each individual protein belonging to the cluster is plotted on top of the centroid. The more intense the colour, the higher the membership score.

**Figure 5 ijms-24-14888-f005:**
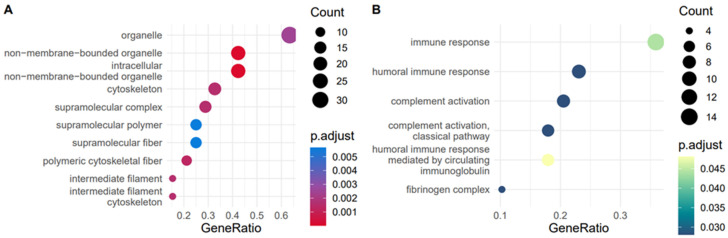
Overrepresentation analysis (ORA) of GO biological process terms enriched in the protein component of multi-omics time series clusters. (**A**) Terms enriched in plasma-derived EVs for cluster 4, (**B**) terms enriched in synovial fluid-derived EVs for cluster 1. The size of the dot represents the number of features that match to that term and the colour represents the adjusted *p*-value.

**Figure 6 ijms-24-14888-f006:**
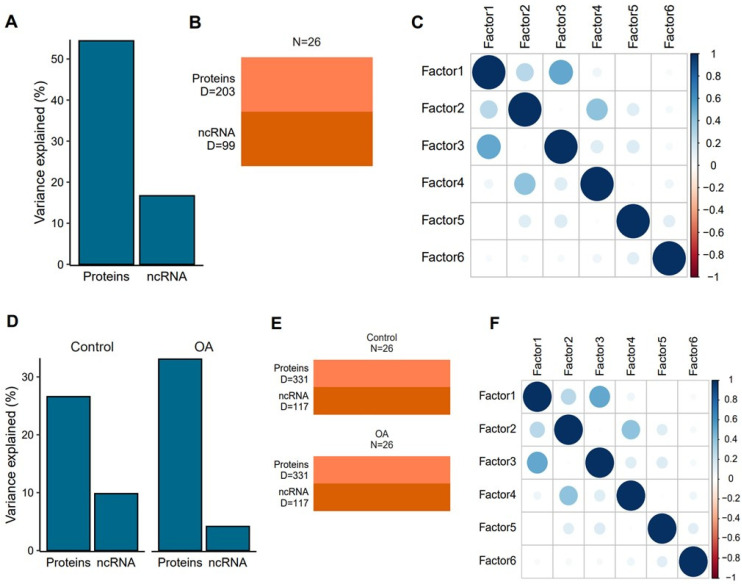
Multi-omics data integration for plasma-derived EVs (**A**–**C**) and synovial fluid-derived EVs (**D**–**F**) using MOFA. Latent factors were identified by the model. (**A**) plasma-EV plot of variance explained per modality, (**B**) number of features per view used to train the model for plasma-EVs, (**C**) Pearson correlation between all factors identified by the model for plasma-EVs. (**D**–**F**) Multi-omics data integration for synovial fluid-derived EVs using the MEFISTO implementation of MOFA2 with multi-group (i.e., “control” and “OA”) comparisons. Latent factors were identified by the model that identify sources of variation within each group. (**D**) Synovial fluid-derived EV plot of the overall variance explained per modality per group, (**E**) number of features per view used to train the model for the control and OA group in synovial fluid-derived EVs, **(F)** Pearson correlation between all factors identified by the model for synovial fluid-derived EVs.

**Figure 7 ijms-24-14888-f007:**
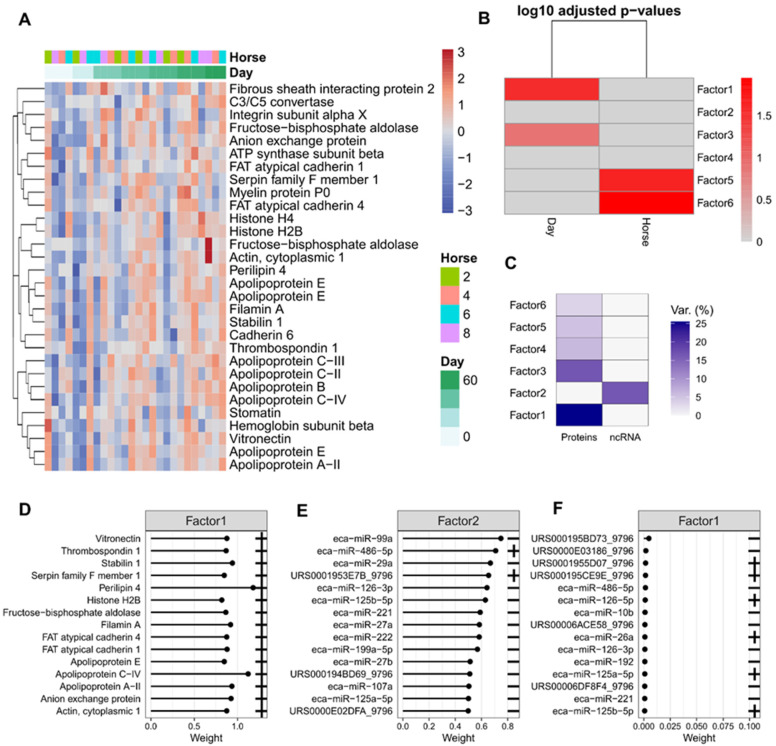
Factor data for plasma-derived EVs. (**A**) Protein heatmap for factor 1, the columns are ordered by day, whereas hierarchical clustering was carried out on the rows. The data were scaled for each feature. (**B**) Pearson correlation between covariates in the phenotype data and latent factors. Colour corresponds to log10 adjusted *p*-values. (**C**) Proportion of total variance (R2) explained by each latent factor per modality. (**D**) Loadings of the top features of Factors 1 in proteomic data (absolute weight). (**E**) Loadings of the top features of Factors 2 for sequencing data. (**F**) Loadings for Factor 1 in sequencing data (absolute weight). Note that the signs are relative within a single latent factor for biomarker weights.

**Figure 8 ijms-24-14888-f008:**
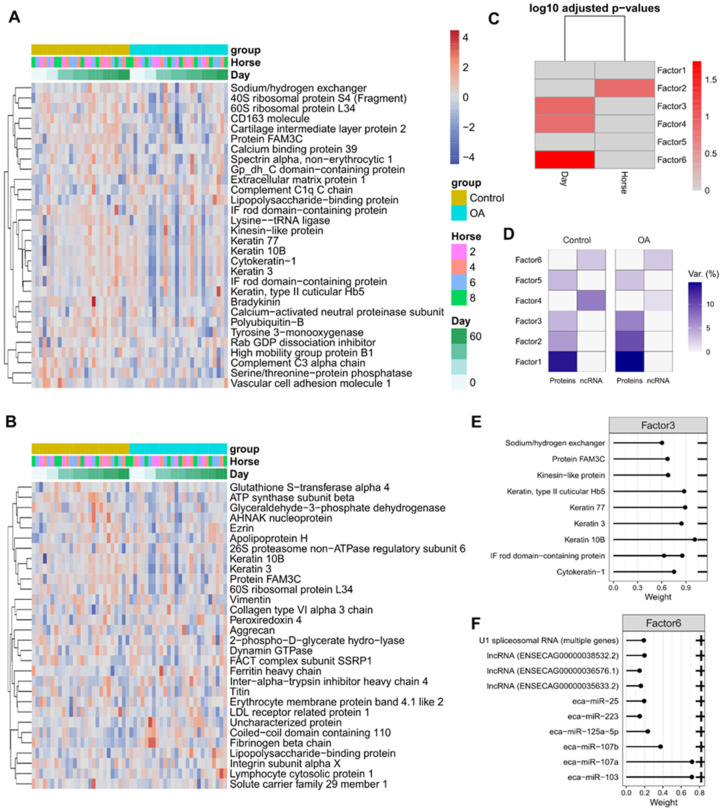
Factor data for synovial fluid-derived EVs. (**A**). Factor 2 protein heatmap. (**B**) Factor 6 protein heatmap. For A and B, the columns are ordered by group then day. Hierarchical clustering was carried out in rows. The data were scaled for each feature. (**C**) Associations between factors and phenotypes, (**D**) proportion of total variance (R2) explained by each latent factor per modality. Absolute weights of the top features of (**E**) Factors 3 proteomics data, (**F**) factor 6 in sequencing data. Note that the signs are relative within a single latent factor for biomarker weights.

**Figure 9 ijms-24-14888-f009:**
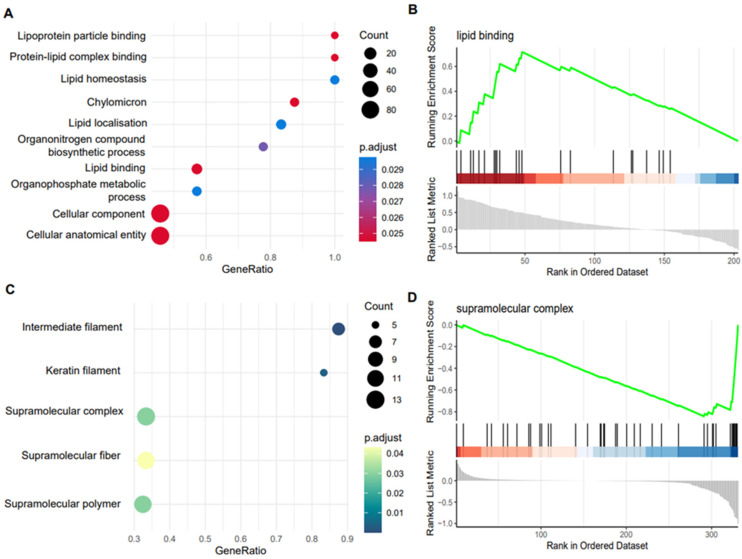
Functional enrichment of latent factors associated with covariates in the phenotype data. Absolute weights of the proteomics data were used as a ranking metric for GSEA analysis. (**A**) Dotplot of plasma-EV factor 1 GSEA results. The size of each dot is proportional to the number of proteins that map to the term. The colour of the dot corresponds to the adjusted *p*-value after correcting for FDR. (**B**) Sample GSEA plot for one of the plasma-EV factor 1 enriched GO terms. The top portion of the plot shows the running enrichment score for the gene set. The middle section shows where the members of the gene set appear in the ranked list. The bottom portion of the plot shows the ranking metric. (**C**) Dotplot of SF-EV factor 3 GO term GSEA results, the interpretation of this is the same as the plasma results above. (**D**) Sample GSEA plot for one of the SF-EV factor 3 enriched GO terms. Only factors 1 for plasma and factor 3 for synovial fluid had significant terms.

**Figure 10 ijms-24-14888-f010:**
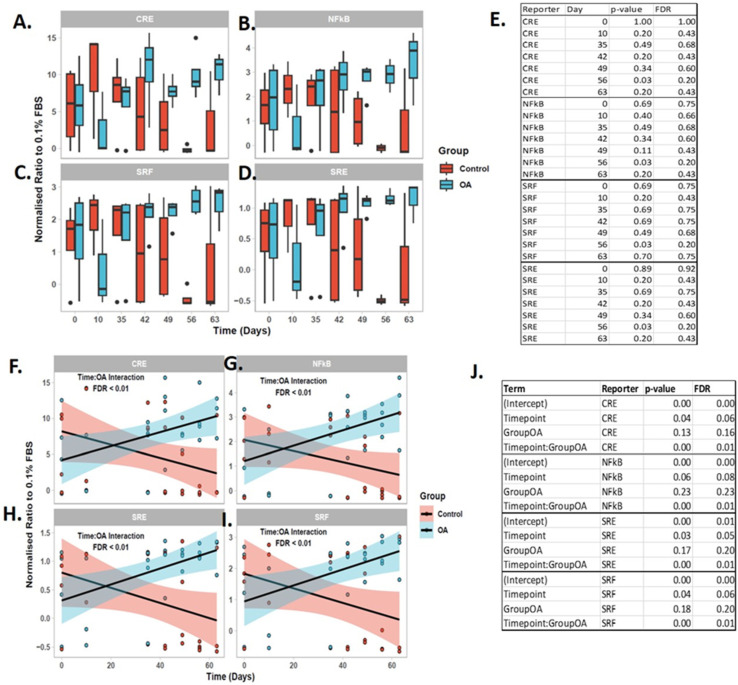
Synovial fluid-derived EV signaling. (**A**) CRE, (**B**) NFKB, (**C**) SRE, (**D**) SRF showed significant changes in reporter activity (false discovery rate < 1%). (**E**). Table of Wilcoxon ranking statistical analysis test result for each time point. Response of transcription reporter assay over time to synovial fluid-EVs for (**F**) CRE, (**G**) NFKB, (**H**) SRE, (**I**) SRF. (**J**) Table of linear interaction output. The models were fitted using a linear interaction model in R using the lm() function. For each reporter with a significant effect, the model was plotted using ggplot2. The fitted model is shown as a line for the OA (blue) and control (red) SF-derived EVs. The 95% confidence intervals for each group are shown as a shaded area. The raw data are included as individual points.

**Table 1 ijms-24-14888-t001:** Number of differentially expressed proteins in plasma and synovial fluid-derived EVs compared to day 0 (*p* < 0.05). Time was either handled as a series of pairwise comparisons against day 0 or as a continuous linear model; this was to identify proteins with linear and non-linear responses to OA induction.

Biofluid	Day (Versus Day 0)	*p*-Value < 0.05
Plasma	10	4
Plasma	35	8
Plasma	42	5
Plasma	49	12
Plasma	56	8
Plasma	63	3
Plasma	continuous	36
Synovial fluid	10	34
Synovial fluid	35	6
Synovial fluid	42	12
Synovial fluid	49	3
Synovial fluid	56	6
Synovial fluid	63	6
Synovial fluid	continuous	2

## Data Availability

Proteomics data are available via ProteomeXchange with identifier PXD041515. Sequencing data presented in this study are available using GEO ID GSE200330.

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
