# Peer review of "Multi-Omic Temporal Landscape of Plasma and Synovial Fluid-Derived Extracellular Vesicles Using an Experimental Model of Equine Osteoarthritis"

_ijms, 2023, doi:10.3390/ijms241914888_

Round 1
Reviewer 1 Report
After careful reading of the manuscript I have to inform you that the bioinformatic analyses are beyond my area of expertise. Please find below comments for the sections I feel confident to review, the rest should be reviewed by a bioinformatician.
Comments:
Section 2.2 – EV characteristics: The reviewer considers this section too scarce. The authors refer to an earlier publication of their group where EV characterisation (which is missing a third characterisation method such as cryoEM as recommended by the MISEV2018 guidelines) was performed using NTA and antibody capture. At least the main findings from the EV characterisation should be mentioned in the present manuscript, it is very annoying for a reader to look this information up in another publication, where the access might be limited as well.
Section 2.3 The term ORA should be explained upon first occurrence in the text and written out. The claim “The cargo of plasma-derived EVs and synovial fluid-derived EVs appears to be unique,..” seems exaggerated, 102 shared proteins is already 50% of all identified plasma EV proteins – what is unique here?
Section 2.4 “mPCA” should be written out and explained upon first occurrence in the text. The acronym is nowhere explained, the reviewer needed to guess that it is some type of principal component analysis. What does a “mild day separation” mean? How is this quantified?
Figure legend 2: What does SWATH-MS mean? Figure 2B and 2C look identical – what is the difference besides different colors?
Table 1: What does “continuous” mean?
Section 2.5: “However, among these proteins Coagulation factor XIII A chain was the only significant protein after correcting for FDR in plasma-derived EVs at day 35.” – Which level of FDR? So this means that there are basically no differences over time for EV-associated proteins neither for plasma nor synovial fluid? The section title of section 2.5 misses information that the paragraph reports about protein data.
Figure 3: It should be indicated in the figure which plot shows plasma or synovial fluid EV data, rather than mentioning this in the legend only.
Section 2.6: How is “a mild separation by time” quantified? What is the noteworthy result that snoRNA U3 and snord58 are differentially expressed, while in the next sentence it is reported that there were no differences whatsoever? And what is the context of the statements to “quality control” as claimed in the section title?
Section 2.7: What was the hypothesis here? The experimental process here seems very unscientific to the reviewer. Apart from noisy plots, no relevant information can be seen. What are the meaning of the results?
Section 4.4. EV Isolation: How long did the hyaluronidase treatment last? It is missing how the purity of EVs and absence of contaminating factor eg from plasma (lipoproteins, macroglobulin, etc) was assessed.
Section 5 Conclusions: The claim “EV-induced changes in signaling responses” was not investigated in the study?
Based on the reviewed sections which are presented in a very confusing way, I recommend a serious major revision in case the bioinformatic data holds relevant key messages.
.
Author Response
Please see attachment for point by point rebuttal.

Reviewer 2 Report
Please see attached file for detailed comments.

There are no spelling mistakes or big grammar mistakes but the paper needs revision by a qualified editor for many small mistakes.
Author Response
Please see the attachment for detailed rebuttal.

Round 2
Reviewer 1 Report
The reviewer thanks the authors to address all comments, one issue still remains:
Line 101-103: The mean and mode sizes of particles were added, however, without error ranges. This is essential as the size distributions of these particles can be broad and/or skewed. By giving just one number for a mean or mode for the individual sample types (control synovial fluid, OA synovial fluid, plasma) creates a misleading perspective that sizes vary depending on sample type. Therefore, please add error ranges for mean and mode sizes and test them statistically for differences to avoid confusion. Presumably there are none and the sizes of the particle populations are very similar.
Author Response
Thank you to the reviewer for their comments, we have now added into the manuscript the standard deviation values and ranges for each of the recorded mean and mode particle sizes, together with the associated statistical analyses. This paragraph now reads as below:
EV characteristics for plasma and synovial fluid-derived EVs at all time points have been reported previously [19]. Briefly, nanoparticle tracking analysis identified the mean and mode particle size (± s.d. and range) for control synovial fluid to be 219.8 (± 49.7, 114.6-360.3) nm and 180.9 (± 44.5, 91.6-319.7) nm, OA synovial fluid 221.8 (± 40.2, 154.9-312.1) nm and 195.0 (± 64.8, 121.7-407.9) nm, and plasma 158.4 (± 18.9, 125.0-203.7) nm and 123.3 (± 22.3, 90.4-163.2) nm. For synovial fluid-derived EVs, no statistical difference in size, size distribution, or concentration between the control and OA joints was identified. However, for plasma-derived EVs, a reduction in the mean EV size was identified on day 49 (147.3, ± 11.2, 133.4-158.2) nm compared to day 0 (167.1, ± 18.5, 144.3-183.0) nm (p = 0.01). Exoview analysis identified positive staining for CD63 and CD81 markers on both synovial fluid and plasma EVs.
Reviewer 2 Report
The authors are to be commended for making all requested changes.
Some minor spelling and other language mistakes present. Please proofread before submission!
Author Response
We thank the reviewer for their assistance and will indeed do this.